# Bending Behavior of Lightweight Wood-Based Sandwich Beams with Auxetic Cellular Core

**DOI:** 10.3390/polym12081723

**Published:** 2020-07-31

**Authors:** Krzysztof Peliński, Jerzy Smardzewski

**Affiliations:** Department of Furniture Design, Faculty of Wood Technology, Poznan University of Life Sciences, Wojska Polskiego 28, 60-637 Poznan, Poland; krzysztof.pelinski@up.poznan.pl

**Keywords:** wood composite, HDF, plywood, cardboard, energy, experiment, FEM

## Abstract

The work concerns a three-point bending test of beams made of plywood, high density fibre boards, cardboard, and wood-epoxy mass. The goal of the investigation was to determine the effect of thickness and type of wood-based facings on stiffness, strength, ability to absorb, and dissipate the energy of sandwich beams with an auxetic core. The cognitive goal of the work was to demonstrate the possibility of using recycled materials for facings and cores instead of popular wood composites. Experimental studies and numerical calculations were performed on correctly calibrated models. Experimental studies have shown that the beams with HDF facings (E = 1528 MPa, MOR = 12.61 MPa) and plywood facings (E = 1248–1395 MPa, MOR = 8.34–10.40 MPa) have the most favourable mechanical properties. Beams with plywood facings also have a good ability to absorb energy (1.380–1.746 J), but, in this respect, the beams manufactured of HDF (2.223 J) exhibited better capacity. The use of an auxetic core and facings of plywood and cardboard significantly reduces the amount of dissipated energy (0.0093 J, 0.0067 J). Therefore, this type of structures can be used for modeling beams carrying high deflections.

## 1. Introduction

Wood-based composites are valued for their technical feasibility, ease with aesthetic configuration, and high strength to density ratio. Their physico-mechanical properties are highly advantageous among other composite materials. In order to achieve low density without deterioration to mechanical properties, new generations of wooden sandwich panels are developed. Research is concerning the development of novel multilayer panels, where core geometry is being modified or made with the use of new material. With the use of well-known, wood-based composites as facings, those novel composites could exhibit superior properties in comparison to traditional materials. Honeycomb, light sandwich panels with a paper core are successfully used in many industries. These materials are commonly used in aeronautics and shipbuilding [1,2,3,4], car industry [5], construction materials, and the furniture industry [6,7,8,9]. Thanks to the periodic structure of the honeycomb core, such composites are often designed with honeycomb and a hexagonal structure that provides satisfactory mechanical properties due to the excellent ratio of relative density to their overall stiffness and strength [10,11,12,13]. Research shows that the overall dimensions of core cells were described by equations that do not include one important factor, which is the cell wall thickness, and treat it as a concise element of the linear wall dimension. This approach is limited and significantly reduces the effect of the cell wall thickness on the analysis of the cell relative density. In the group of the materials of biological origin, the paper is most commonly used in the industrial practice of the cell core fillings industry. Despite shrinking natural resources, bio-fibres are still most often used as the main component, especially considering the paper treatment. Taking this factor into consideration, it encourages the search for new, durable, biobased composites, which could be used as a replacement for conventional honeycomb [14,15,16]. Cellular honeycombs have a wide range of implementation from furniture manufacture to the door industry [17]. Attempts have also been made to find a replacement for the conventional paper core with a corrugated core made out of wood fibres [18,19], plywood, or HDF [20]. Few milestones were achieved in the field of modelling lattice, pyramidal, or auxetic from wood-based composites. Studies focused on the determination of elastic properties of core structures as well as defining the influence of auxetic cell core geometry on the mechanical properties of multilayer panels [16,21]. A finite element method (FEM) with relatively high accuracy could be used to analyze the complex behavior of honeycomb panels such as stress concentration and buckling. This methodology requires more time to build an exact material model that includes the complex nature of the structure and its multiple parameters, especially at preliminary design and optimization [22,23,24,25]. In the multilayer structure of the panels, facings contribute the most to the bending stiffness of the structure, and the main role of honeycomb is to provide out-of-plane support. Moreover, the failure characteristics of experimental specimens are ductile, and the sandwich structure, thanks to the core structure, still has a small amount of bearing capacity and maintains structural integrity [26,27,28,29,30]. A finite element method could be used to analyse the strength of a bonded sandwich beam with a corrugated core and the crack expansion in the multilayer structure [31,32,33].

Taking into consideration the literature studies above, following a research hypothesis was formulated. The type of biomaterial facing significantly affects the ability to absorb and dissipate energy during bending of multilayered beams with an auxetic core. A negative Poisson’s ratio characterized auxetic materials (structures) [8,9,12,13,14,15,16,18]. They improve the mechanical properties of panels, beams, etc., especially shear resistance, bending resistance, impact resistance, and increase of the ability to absorb energy.

The purpose of the work was to determine the effect of thickness and type of wood-based facings on stiffness, strength, ability to absorb, and dissipate the energy of sandwich beams with an auxetic core. The cognitive goal of the study was to demonstrate the possibility of using wood-based materials for facings and cores’ multilayered honeycomb panels. The results of the research were to prove that these materials are excellent for the production of beams susceptible to high deflections.

## 2. Materials and Methods

### 2.1. Tests of Materials

In this work, it was assumed that facings of the multilayered beam would be made out of plywood Ceiba pentandra (L.) Gaertn (PL) by taking into account the longitudinal (L) and perpendicular (R) direction of the fibres of the face veneers, high-density fibreboard (HDF), cardboard (CA), and WoodEpox^®^ composite (WE) as a material of the core structure. The structure of the core is built out of WoodEpox^®^ based on epoxy resin and a lignin-cellulose mass as filler, characterized by zero VOC emissions with the Greenguard^®^ certificate. The choice of plywood and HDF was dictated by their widespread use in the wood industry. The choice of cardboard and WoodEpox^®^ entailed an attempt to use materials that could be admixed with newly manufactured wood-plastic composites as part of the recycling process. Additionally, it was decided to demonstrate the possibility of their use as main materials of layered structures. In particular, large deformations of plywood and cardboard beams could indicate their ability to create structures with synclastic surfaces like the surface of the sphere. Examination of elastic properties and moisture content was held according to the standards [34,35]. In order to determine the contractual elastic limit and the yield strength for the wood composites, uniaxial tensile tests were carried out with use samples, as shown in Figure 1.

An experimental test was performed on a ZWICK 1445 universal strength testing machine (Zwick Roell GmbH & Co.KG, Ulm, Germany). Values of force and deflection were recorded with an accuracy of 0.01 N and 0.01 mm at a frequency of 1 Hz. The stand was illuminated with two LED lamps with a brightness of 630 lumens each. The monochrome photos of the sample before deformation and during loading were taken using the Olympus OM-D camera (Olympus, Tokyo, Japan). The measurements of the strains were made by the edge detection method using the National Instruments IMAQ Vision Builder 6.1 software (National Instruments Poland Sp. z o.o., Warsaw, Poland). The results of uniaxial tensile tests are shown in Figure 2. Based on the stress-strain relationship, the modulus of linear elasticity of facing and core material $E_{f}$ (MPa), $E_{c}$ (MPa), modulus of rupture MOR (MPa), and Poisson’s ratios ϑ were determined for each selected material (Table 1). In Table 1, apart from elastic constants of materials, physical properties were collected including density D (kg/m^3^), absolute moisture content MC (%), and facing thickness $t_{f}$ (mm). Based on the stress-strain dependencies (Figure 2) obtained, the modulus of linear elasticity and modulus of rupture were determined for each selected material (Table 1).

Furthermore, with use of experimental values of stress-strain dependencies, the elastic properties of each material were calibrated using a standard tool of software Abaqus/Explicit v. 6.14 (Dassault Systemes Simulia Corp., Waltham, MA, USA). Abaqus uses the finite element method (FEM) algorithm. The calibration was aimed to fit the ductile damage model to the experimental data and its subsequent application in damage modeling of layered beams.

The experimentally-established stress-strain relationships are characterized by pronounced non-linearity. Therefore, before performing numerical calculations, the beginning of the linear elasticity range, the end of the linear elasticity range, and the end of the plasticity range had to be designated. In order to include plasticity in numerical calculations for selected materials, the experimental stress-strain dependence had to be calibrated. First, the linear elastic range was determined to establish the linear equation for this section. The modulus of linear elasticity $E_{f}$, $E_{c}$ (MPa) (Table 1), true stress, and true strain were described using the equations given below.
(1)$$E_{f},E_{c}=\frac{\sigma }{\varepsilon },$$
(2)$$\sigma_{T}=\sigma \left(1+\varepsilon \right),$$
(3)$$\varepsilon_{T}=\ln\left(1+\varepsilon \right).$$
where: $E_{f}$, $E_{c}$ (MPa)—linear modulus of elasticity for facing and core material, σ (MPa)—engineering stress, ε—engineering strain, $\sigma_{T}$ (MPa)—true stress, $\varepsilon_{T}$—true strain.

The experimental stress-strain curves were calibrated using a standard tool in software Abaqus/Explicit v. 6.14 (Dassault Systemes Simulia Corp., Waltham, MA, USA). The calibration was necessary to estimate true stress and true strain. The converted engineering data obtained from the tensile test for all materials were used to verify their quality. An analogous sample as that in Figure 1 was modeled in the Abaqus. The material damage model was ascribed the elastic modulus and, respectively, determined stresses and strains from the plastic range. Figure 2 also presents stress-strain relationships from experimental and numerical tensile tests for all materials. From this figure, it follows that the damage model used for the selected materials is correctly designed. The differences between the experimental and numerical results are very small. Thanks to the calibration, the ductile damage model was fit to the experimental data of materials and its subsequent application in damage modeling of multilayered honeycomb beams. Detailed model data are presented in the next part of the work describing the numerical model of a multilayered beam.

### 2.2. Manufacturing of Auxetic Wood-Based Sandwich Panels

The entry stage of multilayered honeycomb beams manufacturing was the core preparation. First, the rectangular moulds with the thickness of 12 ± 0.2 mm were filled and stayed for a minimum of 48 h to achieve final stiffness. After hardening, the WE panel was subjected to sanding to even the surface to the final thickness of 10 ± 0.05 mm on a Felder FW 1102 wide belt sander (Felder Group, Żory, Poland). To achieve the designed shape of the core (Figure 3), panels were subjected to CNC milling on a Kongsberg-X milling cutter using an HM straight shank cutter of 3 mm in diameter (CMT, Poznań, Poland) and then cleaned with compressed air. From an industrial point of view, the cores should be produced by extrusion of a WoodEpox^®^.

The shape of a single cell is presented in Figure 4. Relative density $\rho_{A}$ of an analyzed auxetic structure is described as a ratio of core density $\rho_{A}^{*}$ to a density of core substance $\rho_{sA}$:(4)$$\rho_{A}=\frac{\rho_{A}^{*}}{\rho sA}.$$

It could also be described as a ratio of cell surface $F_{sA}$ to the surface of the overall area of the cell $F_{A}^{*}$.
(5)$$\rho_{A}=\frac{F_{sA}}{F_{A}^{*}}.$$

According to Figure 4, the surface of the elementary cell of the core is:(6)$$F_{A}^{*}=L_{x}S_{y}=4\left(t_{A}+l_{A}\cos\left(\varphi_{A}\right)\right)\left(h_{A}-t_{A}\mathrm{ctg}\left(\varepsilon_{A}\right)-l_{A}\sin\left(\varphi_{A}\right)\right),$$

Whereas the surface of the core structure:(7)$$\begin{array}{ll} F_{SA}= & 4\left(t_{A}+l_{A}\cos\left(\varphi_{A}\right)\right)\left(h_{A}-t_{A}\mathrm{ctg}\left(\varepsilon_{A}\right)-l_{A}\sin\left(\varphi_{A}\right)\right)\\ & -4l_{A}\cos\left(\varphi_{A}\right)\left(h_{A}-2t_{A}\mathrm{ctg}\left(\varepsilon_{A}\right)-l_{A}\sin\left(\varphi_{A}\right)\right), \end{array}$$
after necessary transformations:(8)$$\rho_{A}=1-\frac{l_{A}\cos\left(\varphi_{A}\right)\left(h_{A}-2t_{A}\mathrm{ctg}\left(\varepsilon_{A}\right)-l_{A}\sin\left(\varphi_{A}\right)\right)}{\left(t_{A}+l_{A}\cos\left(\varphi_{A}\right)\right)\left(h_{A}-t_{A}\mathrm{ctg}\left(\varepsilon_{A}\right)-l_{A}\sin\left(\varphi_{A}\right)\right)}$$
where $l_{A}$, $h_{A}$, $t_{A}$ (mm)—length, height, and thickness of a cell wall, $\varphi_{A}$ (°)—the angle of inclination of a cell, $\varepsilon_{A}$ (°)—the interior angle of a cell, $S_{y}$, $L_{x}$ (mm)—overall dimensions of the cell, and $l_{0}$ (mm)—distance between cell walls. After calculations, the value of the relative density of the WoodEpox^®^ core reached $\rho_{A}$ = 0.1547.

After cleaning, the core was prepared to glue with selected facings. Facings of sandwich panels were made from PL, HDF, and CA of 3.0, 2.0, and 1.0 mm in thickness, respectively. Facings and cores were glued using PVAC Woodmax FF 12.47 D2 adhesive (Synthos, Oświęcim, Poland) applied at min. 80 g/m^2^ onto surfaces of facings and core. The pressure of 0.03 MPa was applied in the membrane press Artex VPS (Artex, Poland) for 40 min. Then, samples were cut to final dimensions. Samples were prepared to manufacture sandwich panels, as shown in Figure 5. The number and dimensions of the samples are given in Table 2. Samples for three-point bending tests were prepared, according to the EN 310 standard [14]. Orthotropic properties of plywood were included with longitudinal PL(L) and perpendicular PL(R) orientation to the Y axis.

### 2.3. Bending Tests

An experimental test was performed on a ZWICK 1445 universal strength testing machine (Zwick Roell GmbH and Co.KG, Ulm, Germany). In the bending, test samples were subjected to a preliminary load of 50 N and a feed rate of 10 mm/min. Values of force and deflection were recorded with an accuracy of 0.01 N and 0.01 mm at a frequency of 1 Hz. The test was interrupted when 2 mm deflection was reached or at sample failure. In the 3-point bending test, an initial load of 10 N was used. The test was interrupted at sample failure.

Modulus of elasticity E (MPa) for the tested auxetic beams was calculated from the equation below.
(9)$$E=\frac{l^{3}\left(F_{0.4}-F_{0.1}\right)}{4Wt^{3}\left(\delta_{0.4}-\delta_{0.1}\right)}.$$
where $F_{0.1}$ (N), $F_{0.4}$ (N)—the values of forces equal, respectively, 10% and 40% of the maximum force $F_{max}$ (N), $\delta_{0.1}$ (mm), $\delta_{0.4}$ (mm)—deflection values corresponding to the forces $F_{0.1}$ (N) and $F_{0.4}$ (N), t thickness of the sample, l=20t—the distance between supports 240 mm, 280 mm, and 320 mm for CA, HDF, and PL facings, respectively.

Modulus of rupture MOR (MPa) was calculated from the equation below.
(10)$$MOR=\frac{3lF_{max}}{2Wt^{2}}$$

Energy $E_{a}$ (J) absorbed during real three-point bending was calculated with the use of Equation (11), with the graphic function integration F=f(δ) methodology, on the basis of direct experiments.
(11)$$E_{a}=\int_{o}^{\delta }F\left(\delta \right)d\delta$$
where: $E_{a}$ (J)—absorbed energy, *F* (N)—bending force, δ (mm)—deflection.

### 2.4. Numerical Model

It was decided to use numerical solutions that allow rapid assessment of the analysis of unwanted fibre cracks caused by the increase in excessive loads. A popular method of predicting places of material damage is the cohesive zone method (CZM) in the Abaqus system. This method allows the determination of fibre separation during damage. Currently, one of the commonly used failure testing methods in Abaqus is the extended finite element method (xFEM) [36]. This method allows determining crack propagation regardless of the size of the generated finite element mesh. There are many studies [37,38,39,40,41,42,43] on crack propagation processes and the application of the xFEM method. They confirm that the use of this method enables effective determination of critical areas of structures depending on the defined physical processes consistent with real phenomena, which leads to the loss of initial mechanical properties.

In the numerical models of multilayered beams with honeycomb structures (Figure 6), plywood facings were modeled as orthotropic materials whereas HDF, CA, and WE were modeled isotropic materials. Material properties were highlighted in Table 1, where *MOR* also represents the value of maximal principal stress. Thrust and support were made out of steel with modulus of elasticity $E_{s}$ = 2 × 10^5^ MPa and Poisson coefficient $\vartheta_{S}=0.3$. Moreover, between core and facings, additional contacts were modeled with a type of cohesive behavior. The xFEM method was used separately for the core and the facings under the damage analysis. The discretization of the numerical model was performed using the best possible linear hexahedral finite element type C3D8R, with approximately 200,000 mesh elements and approximately 100,000 nodes. The eight-node elements with three degrees of freedom and reduced integration (C3D8R) are characterized by a very high degree of accuracy of the results. This was obtained by the application of higher-order polynomial equations to describe this process [44]. The phenomenon of thermoplasticity was not included in the calculations. In case of the glue line, this phenomenon would be noticeable at temperatures above 90 °C or variable higher (over 100 mm/min) load speed. The laboratory conditions in which the test was carried out were constant air humidity 60%, constant air temperature of 21 °C, and constant load speed of 10 mm/min.

Computations were performed at the Poznań Supercomputing and Networking Center (PSNC) using the Eagle computing cluster and Abaqus (Dynamic, Explicit) v.6.14 (Dassault Systemes Simulia Corp., Waltham, MA, USA).

## 3. Results and Discussion

### 3.1. Mechanical Properties of Multilayered Beams

Analyzing the results of experiments, a significant effect of different facing material on the value of E and MOR is observed. The difference in the *E* values is significant when comparing samples with HDF (1528 MPa), PL(L) (1395 MPa), or PL(R) (1248 MPa) facings to the CA (117 MPa) combination (Figure 7). In relation to beams with facings manufactured using HDF, the modulus of linear elasticity of beams from PL(L), PL(R), and CA is lower by 9%, 18%, and 92%, respectively. A similar tendency is shown by the relationships between the modulus of rupture. In relation to beams with facings manufactured by HDF, the MOR of beams from PL(L), PL(R), and CA are lower by 18%, 34%, and 81%, respectively.

The presented relations are clear in the context of the relationship between the modulus of linear elasticity of facing materials and their moments of inertia. From Table 1, it is seen that HDF have a linear modulus of elasticity E = 6880 MPa. Comparing with HDF, facing materials PL, CA, and core material WE have lower linear modulus of elasticity by 6%, 68%, and 84%, respectively. It results directly from the equation below.
(12)$$EI=\sum_{i=1}^{n}E_{i}I_{i},$$
were I (mm^4^)—a moment of inertia of beam, $E_{i}$ (MPa)—modulus of elasticity of single layer, $I_{i}$ (mm^4^)—moment of inertia of a single layer, hence:(13)$$\frac{1}{12}EWt^{3}=\frac{1}{12}E_{c}Wt_{c}^{3}+2E_{f}\left(\frac{1}{12}Wt_{f}^{3}+Wt_{f}{\left(\frac{t_{c}+t_{f}}{2}\right)}^{2}\right),$$
(14)$$E=\frac{1}{t^{3}}\left(E_{c}t_{c}^{3}+2E_{f}t_{f}^{3}+6E_{f}t_{f}t^{2}\right).$$

Excluding the common factors such as $E_{f}$, $t_{f}$, and t before the parenthesis, we can evaluate the effect of individual parameters on the value of the modulus of linear elasticity of the beam.
(15)$$E=2E_{f}{\left(\frac{t_{f}}{t}\right)}^{3}\left(1+\frac{1}{2}\frac{E_{c}}{E_{f}}{\left(\frac{t_{c}}{t_{f}}\right)}^{3}+3{\left(\frac{t}{t_{f}}\right)}^{2}\right).$$

By entering into Equation (15), the appropriate numerical values from Table 1, the impact of the selected parameters on module E was calculated for cardboard facings (CA):(16)$$E=2E_{f}\cdot 0.083^{3}\left(1+\frac{1}{2}\times 0.5157\times 1000+3\times 144\right),$$
for plywood (PL(L)):(17)$$E=2E_{f}\cdot 0.19^{3}\left(1+\frac{1}{2}\times 0.174\times 37.04+3\times 85.33\right).$$

The results show a warning that the linear modulus of elasticity of the beam was determined by the quotient $t_{f}/t$ and the inverse of this quotient. If a 1-mm thick cardboard is used, the quotient is equal to 0.083. For plywood 3-mm thick, the quotient is equal to, respectively, 0.19. It follows the rule that thicker facing in multilayered beams has higher rigidity. It can also be confirmed by the inverse of this quotient. For cardboard, the quotient $t/t_{f}$ is equal to 144, while, for plywood, the quotient is equal to 85. The ratio of core thickness to the facing thickness is also important $t_{c}/t_{f}$. For analogous materials shown above, the relevant quotients are equal to 1000 and 37.04. This means that thicker cores enhance the beam rigidity. Undoubtedly, the influence of materials used to manufacture cores and facings is also significant. This illustrates the quotient $E_{c}/E_{f}$. For cardboard facings, it is equal to 0.5157, and, for plywood, it is equal to 0.174. This convinces us that a higher modulus of linear elasticity of facings significantly improves beam stiffness.

The influence of the elastic properties of facings on the elastic properties of sandwich panels was analytically confirmed in the works [45,46]. The same relationships were also confirmed in three-point bending tests of isotropic cell plates with an aluminium core with hexagonal cells [47] and sandwich composite structures reinforced by basalt fibre and Nomex honeycomb [26]. Similar relationships were also observed during bending of laminated beams made of wood materials in which the cladding was reinforced with glass fibre [48] or laminated paper [22].

In the case of multilayered structures, it is preferable to compare linear elasticity modulus and modulus of rupture in relation to their density [14]. To this moment, two coefficients have been determined as:(18)$$Q_{E}=\frac{E}{\rho },$$
and
(19)$$Q_{MOR}=\frac{MOR}{\rho }.$$
where: $Q_{E}$ and $Q_{MOR}$ (Nm/kg)—respectively, are the coefficient of relative rigidity and the coefficient of relative strength, *ρ* (kg/m^3^)—density.

In Figure 8, it is shown that the best combination was obtained for panels with PL (L) facings ($Q_{E}$ = 4.620 Nm/kg). The smallest $Q_{E}$ values were obtained for beams with CA facings (0.516 Nm/kg). A high $Q_{E}$ factor indicates that the composite produced is light and highly rigid. Compared to HDF beams, multilayer structures with PL have a better $Q_{E}$ factor of about 8.0% to 23.5%. Slight differences in regularities were noted for the coefficient $Q_{MOR}$. The best properties were shown by beams with PL(L) facing (Figure 8). The structure with cardboard facings turned out to be the least attractive.

### 3.2. Damage and Energy Absorption by the Sandwich Beams

Figure 9 shows the Mises stress distribution caused by three-point bending of beams. It follows that the largest stresses are located in the impact zone. In the bottom, stretched facing, the stresses are 22.9, 19.5, 13.8, and 11.3 MPa for PL (L), PL (R), HDF, and CA, respectively. These values correspond to the strength of individual materials listed in Table 1, which indicates the accuracy of the developed numerical models and the effectiveness of further calculations. Figure 10a also confirms the accuracy of calibrating the numerical model. Experimental and numerical values of deflection of beams subjected to bending are included in it. It is particularly interesting that, at the load peak, each curve illustrating the result of numerical calculations fits well with the experimental curve. Panels with HDF facings are characterized by the highest values. For this type of material, the highest damaging force was noted at the point of 305 N. Panels with HDF facings are characterized previously in this field. For this type of material, the highest damaging force was noted at the point of 305 N. Beams with PL(L) facings are characterized by the relatively high value of damaging force 225 N, but beams with PL(R) facings exhibited values as low as 51 N, which means a reduction by 77%. For the CA beams, values of load at the damage point where the lowest among all tested beams equal 42 N. However, this type of material exhibits the best correlation between FEM simulation and experiment. The values of the FEM and experiment are satisfying for HDF, PL (L), and CA facings and for the end areas of the curve, which did not exceed 5% of the difference. However, for the PL(R) sandwich panel, the difference is significant and suggest further experiments to properly simulate material behavior.

The xFEM method allowed us to determine the places of damage in the core and facings. The value of a horizontal displacement at the point of the formation of the crack is considered as a measure of these damages. The relationship of horizontal displacement as a function of sample deflection during bending is shown in Figure 10b. The highest values of horizontal displacement were noted for PL(L) multilayered beams (0.35 mm). The lowest value was exhibited by beams with CA facings (0.15 mm). For the beams with PL(R) and HDF facings, horizontal displacement is very close at, respectively, 0.25 and 0.20 mm. The highest value of the crack for beams PL(L) with plywood oriented longitudinally to the beam axis are also caused by its internal structure where the crack is initiated by the fibre fracture. HDF and CA are homogenous materials. Therefore, the values of the horizontal displacements are relatively lower.

The significant effect of type of facings material on mechanical properties of multilayered beams could be noted for the crack occurrence of the facings and core structure of tested beams (Figure 11). For the plywood PL(L), the main crack occurs in the facing material. In this case, the highest local value of the reduced Mises stresses is equal to 32 MPa (Figure 11a). Visible cracks in the cell walls of the core are caused by small stresses not exceeding 3 MPa. For the PL (R) beams, local damage stresses are equal to 26.2 MPa (Figure 11b). Cracks in facings made of HDF are starting to occur at stresses of approximately 9.2 MPa, while, in the case of cardboard facings, occur at stresses of approximately 8.3 MPa (Figure 11c,d). It should be noted that the damage appeared mainly in the bottom facings. When the above-mentioned local reduced stresses were exceeded, the facings cracked. As a result of these cracks and the widening of the gaps, there were cracks in the cell walls of the core. This was due to the dominant presence of normal stresses and a small share of tangential stresses. Therefore, the impact of shear stresses on core damage was small. As long as there was no crack in the facings, no cracks in the core were observed.

The change of Mises average stress values into the cracks is illustrated in Figure 12. It shows that the stresses, as a function of deflection, in the vast majority, increase nonlinearly and digressively. For particular types of facings PL(L), PL(R), HDF, and CA, they reach the maximum value of 12.3, 9.82, 5.94, and 4.1 MPa, respectively.

In Figure 13a, it is shown that all beams exhibit the progressive capacity to absorb energy during bending. At the same time, it can be seen that the results of numerical calculations correspond well to the values determined on the basis of experimental tests. When using plywood as facings, the difference between the numerically calculated result and the experiment varies from 17.1% to 22.4%, and, in the case of HDF and CA, varies between 2.9% and 5.1%, respectively (Table 3).

The best capacity for energy absorption is exhibited by beams with HDF facings (2.223 J). However, plywood beams PL(L) and PL(R) also exhibit relatively good energy absorption properties, respectively, 1.746 J and 1.380 J. In case of longitudinal orientation of fibres, the PL(L) absorb 21% energy more than PL(R) when fibres are oriented perpendicularly to the long axis of the beam. This regularity corresponds well to the relationship between higher modulus of elasticity for PL(L) and less modulus of elasticity for PL(R). Beams with CA facings exhibit a significantly lower capacity to energy absorption (0.39 J). In comparison with HDF, those type of facings absorb 82% energy less. Figure 13b also shows the relationship between dissipation energy as a function of beam deflection. During bending of layered beams, deformations at the micro-level are becoming irreversible plastic deformations, which are associated with energy dissipation. This is considered the main factor for causing damage to the material and the formation of fatigue microcracks. Recordings of dissipation energy changes can be a source of information about the occurrence of damage. A more accurate diagnosis of this phenomenon may, therefore, have diagnostic significance. The beams with PL (L) facings are characterized by the highest value of dissipated energy (0.035 J). The beginning of the energy dissipation process occurs at a deflection of about 6 mm. The value of dissipated energy for the rest of the materials did not exceed 0.015 J, and the beginning of dissipation was revealed at the deflection of 10 mm. On this basis, it can be shown that beams made out of HDF (0.0157 J) and especially PL(R) (0.0093 J) or CA (0.0067 J) exhibited less susceptibility to damage when compared to PL(L). Considering this, we can suggest using PL(R) and CA to create layered cell panels with a synclastic shape structure.

The use of a cell core with auxetic properties was of significant importance in this work. Auxetic core cells exhibit a different behavior compared to traditional hexagonal cells. As described in the works [12,13,14,16], the auxetics increase their width during tension and decrease the width under compression. From this reason, close to the bottom facings, the core was tensiled and minimally expanded. However, close to the top facings, the core was compressed and slightly decreased in width. These phenomena do not exist in conventional honeycomb panels. Therefore, auxetic cores enable the formation of honeycomb panels with synclastic surfaces.

## 4. Conclusions

Experimental studies have shown a significant impact on the type of wood-based composites used as facings on the rigidity and strength of beams with an auxetic core. Beams with HDF facings (E = 1528 MPa, MOR = 12.61 MPa) and plywood facings (E = 1248–1395 MP, MOR = 8.34–10.40 MPa) have the most favorable mechanical properties. Considering the relationship between the modulus of linear elasticity or modulus of ruptures and the density of materials, the most advantageous structure is a beam with PL(L) plywood facings. Beams with plywood facings also have a good ability to absorb energy (1.380–1.746 J), but, in this respect, the beams manufactured of HDF (2.223 J) exhibit better capabilities. The use of an auxetic core and facings of HDF, PL (R), and CA significantly reduces the amount of dissipated energy 0.0157 J, 0.0093 J, and 0.0067 J. Therefore, this type of structure can be used to model beams carrying high deflections.

## Figures and Tables

**Figure 1 polymers-12-01723-f001:**
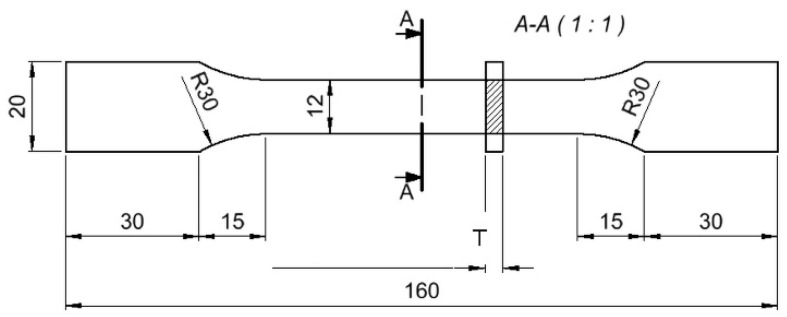
Sample for uniaxial tensile testing of selected materials (dimensions in mm).

**Figure 2 polymers-12-01723-f002:**
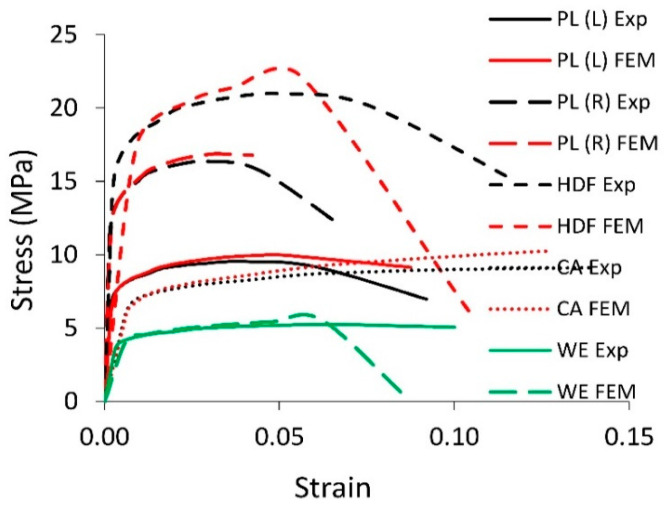
The stress-strain relationship for selected materials. Codes in Table 1.

**Figure 3 polymers-12-01723-f003:**
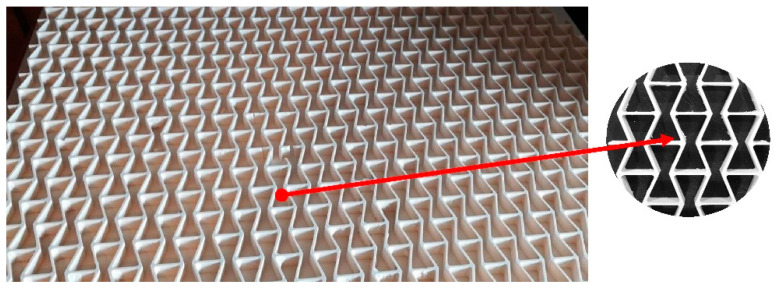
The core with auxetic re-entrant cells.

**Figure 4 polymers-12-01723-f004:**
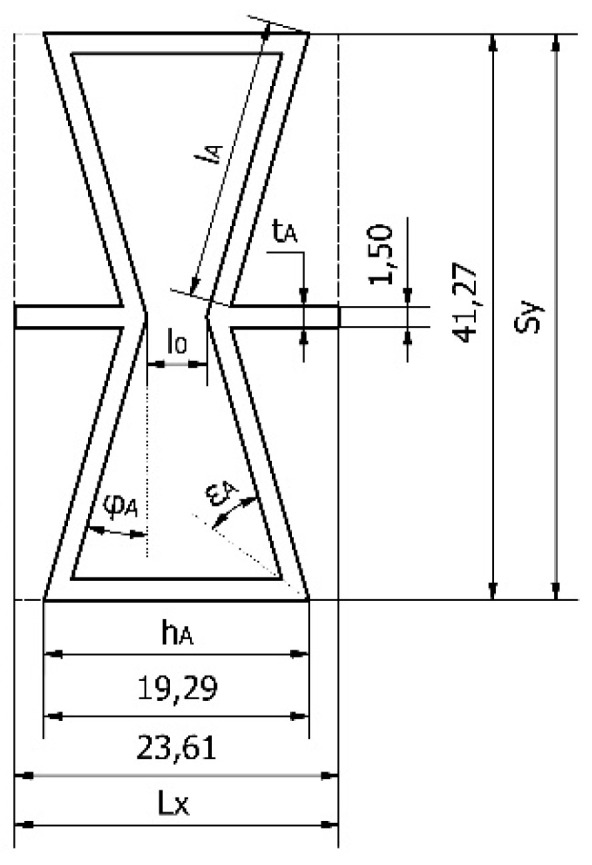
An elementary cut of the auxetic core structure (dimensions in mm).

**Figure 5 polymers-12-01723-f005:**
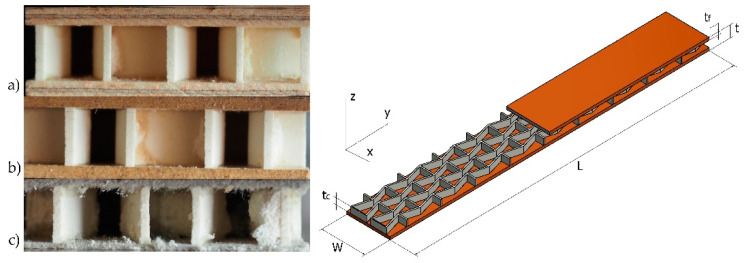
Cross-sections of the beam sample: (**a**) PL, (**b**) HDF, (**c**) CA, $t_{f}$, $t_{c}$, t—thickness of facing, core and board, W,L—width and length of the sample.

**Figure 6 polymers-12-01723-f006:**
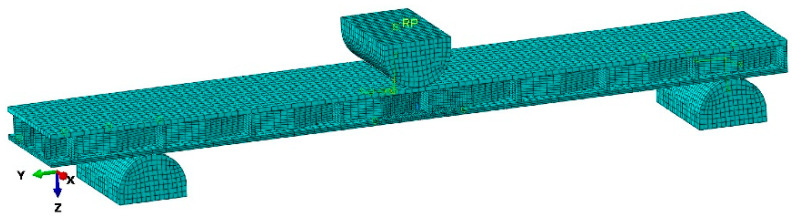
Mesh model of the multilayered beam.

**Figure 7 polymers-12-01723-f007:**
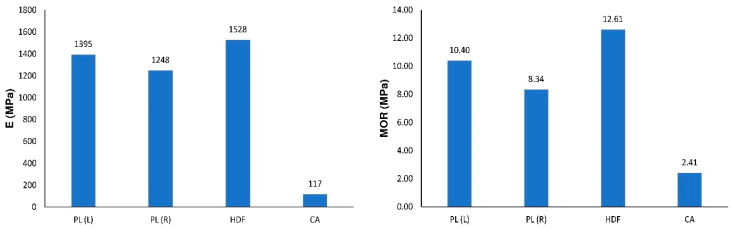
A comparison of E and MOR for selected wood-based composites.

**Figure 8 polymers-12-01723-f008:**
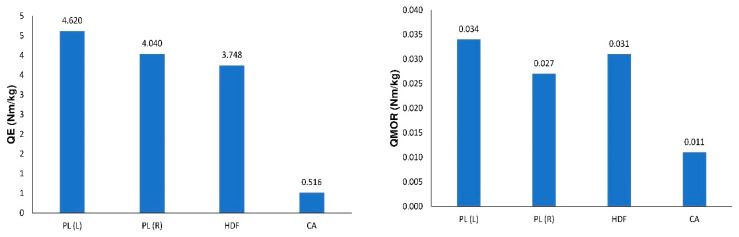
A comparison of $Q_{E}$ and $Q_{MOR}$ for selected wood-based composites.

**Figure 9 polymers-12-01723-f009:**
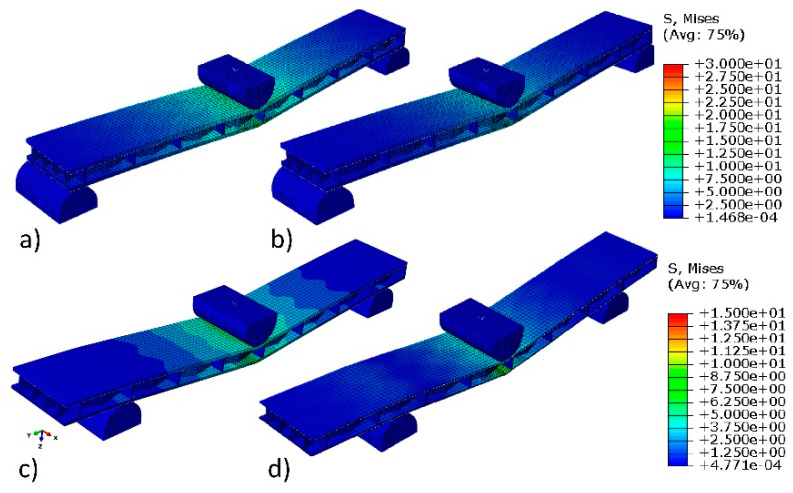
Examples of beams’ deflection: (**a**) PL(L), (**b**) PL(R), (**c**) HDF, and (**d**) CA.

**Figure 10 polymers-12-01723-f010:**
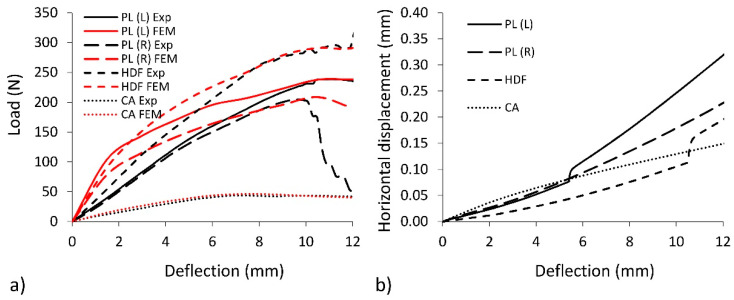
Stiffness characteristics of multilayer beams and (**a**) value of numerical calculations of crack displacement during bending (**b**).

**Figure 11 polymers-12-01723-f011:**
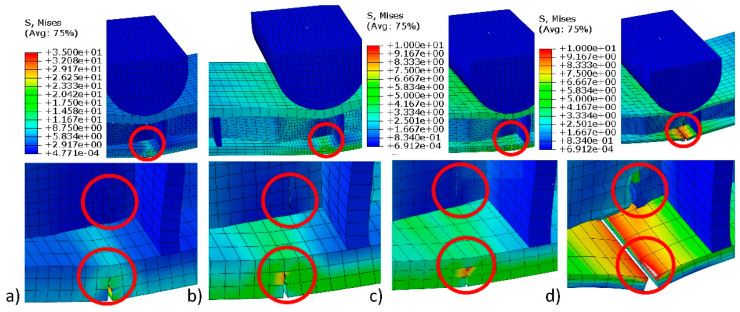
Examples of the cracks in the facings and core of multilayer beams: (**a**) PL(L), (**b**) PL(R), (**c**) HDF, and (**d**) CA.

**Figure 12 polymers-12-01723-f012:**
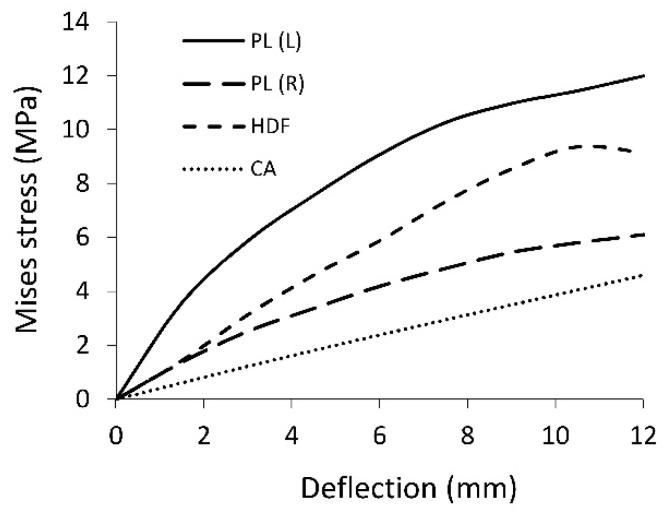
Mises stress at facing cracking point.

**Figure 13 polymers-12-01723-f013:**
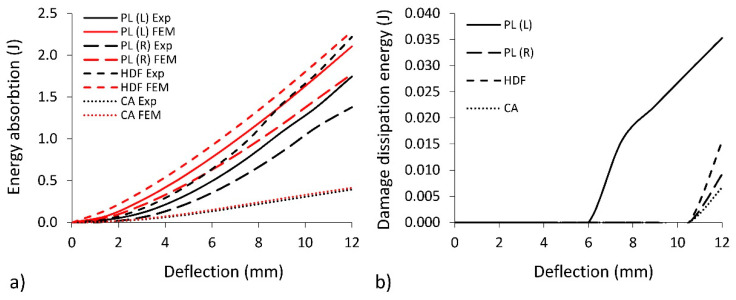
The change in the absorbed energy (**a**) and dissipation of damage energy (**b**) at different deflections.

**Table 1 polymers-12-01723-t001:** Physic-mechanical properties of selected materials [9].

Material	Code	Statistic	*t_f_* (mm)	MC (%)	*D* (kg/m^3^)	*E_f_* or *E_c_* (MPa)	*MOR* (MPa)	ϑ (-)
Superform^®^ Plywood L	PL(L)	Mean	3.10	6.15	350	6473	23.5	0.35
		SD	0.15	0.20	7	125	2.5	0.02
Superform^®^ Plywood R	PL(R)	Mean	3.05	6.10	350	4874	11.4	0.33
		SD	0.15	0.15	9	80	1.2	0.01
HDF	HDF	Mean	2.05	5.90	870	6880	20.5	0.30
		SD	0.10	0.15	8	214	2.1	0.01
Cardboard	CA	Mean	1.02	5.84	615	2170	12.0	0.35
		SD	0.06	0.12	10	108	1.2	0.03
Abatron WoodEpox^®^	WE	Mean	4.51	6.12	430	1119	6.9	0.40
		SD	0.03	0.06	12	49	0.6	0.02
PVAc		Mean	0.10	-	-	460	-	0.30
		SD	-	-	-	-	-	-

**Table 2 polymers-12-01723-t002:** Dimensions of samples.

Sample Type	t (mm)	WxL (mm)	Number of Samples
PL(L)	16	50 × 370	20
PL(R)	16	50 × 370	20
HDF	14	50 × 330	20
CA	12	50 × 290	20

**Table 3 polymers-12-01723-t003:** Energy absorbed and dissipated during bending (standard deviation in parentheses).

Facing Type	Energy Absorbed (J)		Differences (%)	Energy Dissipated (J)
	Exp	FEM		
PL(L)	1.746 (0.105)	2.106	17.1	0.0353
PL(R)	1.380 (0.097)	1.777	22.4	0.0093
HDF	2.223 (0.111)	2.288	2.9	0.0157
CA	0.395 (0.032)	0.416	5.1	0.0067

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
