# Peer review of "Bending Behavior of Lightweight Wood-Based Sandwich Beams with Auxetic Cellular Core"

_polymers, 2020, doi:10.3390/polym12081723_

Round 1

Reviewer 1 Report

The paper described an interesting work on experimentation as well as simulation of the mechanical properties of the sandwich beam with an auxetic core, made of different types of wood. However, except for the use of different types of wood as can be considered bio-materials, I have not recognized any data concerned or deep analyzed about bio-field. The finding of this work is related to the mechanical behavior of the building materials from an overall perspective. Here authors can find some guidelines for the improvement of their manuscript:

  1. The last paragraph in the introduction section describes the detail of the work and unessential to include in the manuscript.
  2. The determination of MOR, Poisson ratio in table 1 should be clarified.
  3. Please explain further about the behavior of the auxetic cellular core under the Test condition or under the Service condition for clarifying the role of this part in the sandwich beam compared to the common ones.
  4. There is a significant effect of different facing material on the value of mechanical properties like E and MOR, please compare the results found herein with respect to those in the literature or those in the market.

Author Response

Responses to the Editor and the Reviewers

Dear Editor and the Reviewers,

First of all, we want to thank for offering the authors the opportunity to respond to the reviewers comments. Below, step by step, we present our explanations and changes introduced to the text. All changes (in manuscript) are marked in red.

Responses to Reviewer #1:

We are grateful to the Reviewer for careful reading the manuscript and helpful remarks. Below we respond to the remarks one by one.

     1. The paper described an interesting work on experimentation as well as simulation of the mechanical properties of the sandwich beam with an auxetic core, made of different types of wood. However, except for the use of different types of wood as can be considered bio-materials, I have not recognized any data concerned or deep analyzed about bio-field. The finding of this work is related to the mechanical behavior of the building materials from an overall perspective. Here authors can find some guidelines for the improvement of their manuscript:

A: The work does not describe the use of wood but wood-based materials and their influence on the properties of bent beams with an auxetic core. A clear emphasis on this aspect of the study of wood based composites was put in the research hypothesis and goal of the work.

     2. The last paragraph in the introduction section describes the detail of the work and unessential to include in       the manuscript.

A: Very often, the editors of the scientific journals accept this type of introduction structure to bring the readers closer to the scope of the work. However, as suggested by the Reviewer #1, this fragment of the text was omitted.

     3. The determination of MOR, Poisson ratio in table 1 should be clarified.

A: The term of MOR and Poisson ratio was described in the text (line 103, 104).

     4. Please explain further about the behavior of the auxetic cellular core under the Test condition or under the Service condition for clarifying the role of this part in the sandwich beam compared to the common ones.

A: The use of a cell core with auxetic properties was of significant importance in this work. Auxetic core cells exhibit a different behavior compared to traditional hexagonal cells. As described in the works [12-14,16], the auxetics increase their width during tension and decrease the width under compression. From this reason, close to the bottom facings, the core was tensiled and minimally expanded. But close to the top facings, the core was compressed and slightly decreased in width. These phenomena do not exist in conventional honeycomb panels. Therefore, auxetic cores enable the formation of honeycomb panels with synclastic surfaces.

     5. There is a significant effect of different facing material on the value of mechanical properties like E and MOR, please compare the results found herein with respect to those in the literature or those in the market.

A: The influence of the elastic properties of facings on the elastic properties of sandwich panels was analytically confirmed in the works [45, 46]. The same relationships were also confirmed in three-point bending tests of isotropic cell plates with an aluminium core with hexagonal cells [47] and sandwich composite structures reinforced by basalt fibre and Nomex honeycomb [48]. Similar relationships were also observed during bending of laminated beams made of wood materials, the cladding of which was reinforced with glass fibre [49] or laminated paper [50].

Jerzy Smardzewski

Reviewer 2 Report

See attached paper.

Author Response

Responses to the Editor and the Reviewers

Dear Editor and the Reviewers,

First of all, we want to thank for offering the authors the opportunity to respond to the reviewers comments. Below, step by step, we present our explanations and changes introduced to the text. All changes (in manuscript) are marked in red.

Responses to Reviewer #2:

We are grateful to the Reviewer for careful reading the manuscript and helpful remarks. Below we respond to the remarks one by one.

  1. What means satisfactory? Do you mean "low density"?

A: Yes, "satisfactory" it means low density.

  1. What means wood fibers plywood structures?

A: This part of text described a corrugated core made out of wood fibres [18,19], plywood or HDF [20]. It is not the wood fibres plywood structures.

  1. The term "auxetic core" must be explained.

A: The explanation was added to the text.

  1. What is the influence of the "auxetic core" in comparison to standard or traditional cores? Why is laid so much importance on the "auxetic core", whereas on the other side the importance of the facings is described?

A: The description of the main properties of auxetic materials (structures) has been added in the introduction. However, this study did not emphasize the special influence of the auxetic core on the test results. The type of core is a constant factor in the research. In line with the aim of the study, only the influence of facings type on mechanical properties of beams was investigated. Therefore, at work, little attention is paid to the auxetic core. But, it was necessary to be emphasized, however, that the core is a cellular and auxetic structure, it means with a negative Poisson's ratio. Thus, the beam used is different from traditional sandwich beams with a honeycomb core.

  1. "Recycling materials" should be defined here.

A: The term "recycling materials" was replaced by biomaterials.

  1. The work …

A: The cognitive goal of the study …

  1. "several" instead of "few"

A: Yes.

  1. Has been also cardboard in the core used, as can be understood from the sentence in lines 90-91? But this is then different information to line 87-88, where only WoodEpox is mentioned as core material.

A: The sentence has been improved to the form: "Also, it was decided to demonstrate the possibility of their use as main materials of layered structures".

  1. What means "synclastic" surfaces? The reader might not know this.

A: In particular, large deformations of plywood and cardboard beams could indicate their ability to create structures with synclastic surfaces like the surface of the sphere.

  1. Please explain more in detail the composition of this material. Is epoxy resin a part of this material, as the name indicates? Which natural resins are used? According to the website of Abatron it is 100% epoxy resin.

A: The description of WoodEpox® presented in the manuscript is a full description of the manufacturer, Abatron company. The manufacturer does not specify the chemical composition of this product but informs that it is a mixture of wood particles and 100% epoxy resin. Using this fact in the manuscript authors write that "The structure of the core is built out of WoodEpox® based on lignin-cellulose mass and resins, characterized by zero VOC emissions with the Greenguard® certificate", according to the best knowledge of the manufacturer.

  1. The codes for the various curves must be explained somewhere or in the captions of Fig. 2. It might be also explained in Table 1 with relevant indication to this table in the captions of Figure 2.

A: The codes for the various curves were explained in the Table 1 with relevant indication to this table in the captions of Figure 2.

  1. … fibres of the face veneers …

A: The sentence has been completed.

  1. Why do you use "ln" in the formula?

A: The equation 3 describe true strain  so, (ln) is necessary.

  1. What is ε'? Or is this just a comma in the sentence?

A: Yes, of course, it is a comma in the sentence. 

  1. There are again commas, where it is not clear if this is part of the formula or really only a comma. Please eliminate, if it is really just a comma in the sentence.

A: Yes, of course, it is a comma or dot at the end of the sentence. 

  1. When calculating the relative density out of the cross section area of the cells you have to take the whole area Lx*Sy as basis, not the area which is enclosed by the cell.

A: Yes, that's what equation 5 is about. Equation 6 additionally shows that the area Lx * Sy is a function of the dimensions of the cell walls. During designing a honeycomb cell, we first describe its geometry using the dimensions of the walls. Thus, Lx and Sy are only the results of this modeling. 

  1. How is defined ? Is it identical to ?

A: No,  is not identical to . The  (°) it is the angle of inclination of a cell,  (°) it is the interior angle of a cell. It was shown in Figure 4 and defined in equations 6,7,8. 

  1. How far is the thermoplastic behaviour of the PVAc considered in all calculations? Especially at long term loading PVAc bond lines tend to some creep. The low Ec or Ef show the big difference to all other materials used.

A: The phenomenon of thermoplasticity mentioned by the Reviewer #2 does not occur in the laboratory conditions in which the test was carried out (constant air humidity 60%, constant air temperature 21°C and constant load speed 10 mm/min). The phenomenon of the thermoplastic behavior of the glue line would be noticeable at temperatures above 90°C or at variable, higher (over 100 mm / min) load speed. For these reasons, thermoplasticity was not included in the calculations.

  1. Uniaxial compression, what does this mean in connection with a bending test?

A: The sentence was improved. ”In the bending, test samples were subjected to a preliminary load of 50 N and feed rate of 10 mm/min”.

  1. When measuring MOR, is there also and evaluation of the type of fatal crack, means e.g. tension crack in the lower face materials (which is under tension load), or shear collapse of the bond line between core and face layers, or buckling of the cell walls of the core? How is this considered in the results?

A: The experimental MOR measurement was carried out in accordance with EN 310. So, tension crack in the lower face materials or shear collapse of the glue line between core and face layers, or buckling of the cell walls of the core were not observed and detected.

  1. With a type of cohesive behavior. What does this mean? 

A: The contact between the parts of the FEM numerical model can be bonded (tie), friction or cohesive. The use of a cohesive contact allows the modeling a glue line between the core and the facings.

  1. The correct numbers for E are -9, -18, and -92%. For MOR: -18, -34, and -81%. Full digit numbers is accurate enough. Minus more than 100% would give a negative value which is physically not possible. Basis for reduction in % is the highest value measured. 

A: Thank you for your apt suggestion. The corrections have been incorporated in the text.

  1. It is necessary to give a comment how such core material should be manufactured in industrial scale. Preparing a plate and then cutting the cell structure out of this plate is much too expensive. 

A: Yes, preparing a plate and then cutting the cell structure out of this plate is much too expensive. From an industrial point of view, the cores should be produced by extrusion a WoodEpox (R). We are constantly looking for an industrial partner to implement this idea. Short information was introduced in the text regarding the planned manufacturing technology. 

  1. It might be better to replace the letter J with I, because J is already in use as dimension for energy as e.g. in line 234 and others. 

A: Yes, thank you for the suggestion. 

  1. It As above: reduction cannot be higher than 100%, otherwise it would be a negative number. The correct numbers are: -6, -68, and -84% 

A: Thank you for your apt suggestion. The corrections have been incorporated in the text. 

  1. 85 

A: 85.33 – it is correct. See equation 17. 

  1. differences 

A: Ok. 

  1. 306; no digit after the comma 

A: Ok. 

  1. Why is there a minus sign? 

A: It was corrected. 

  1. This would be then a negative number. Better: reduced by 77% 

A: It was corrected. 

  1. This is the only case where the fracture behaviour was mentioned. Please describe in detail the fracture behaviour of the various combinations, see also comment further up. 

A: This aspect was explained in answer 39. 

  1. Do the calculated fracture zones correspond with the fracture zones seen in the practical tests? 

A: Unfortunately, the cracks in the laboratory samples were random, but always close to the numerically designated crack. 

  1. This needs to be explained better; the numbers of the displacement are quite small, so how can they be measured? Or are they only calculated/simulated? Is it possible to compare simulated results and practically measured results? 

A: Figure 10a shows the fit of the experimental results and numerical calculations in the form of the load-displacement relationship. They were found that results are sufficiently compatible to discuss the amount of energy absorbed during bending based solely on numerical calculations. Therefore, horizontal displacement in cracks was not measured during the real test. These results come from numerical calculations only. It was described in the text and in the caption under Figure 10b. 

  1. Values for reduction higher than 100% are physically nonsense. Please recalculate. 

A: Thank you for your apt suggestion. The corrections have been incorporated in the text. 

  1. How are the absorbed and the dissipated energy calculated? How is the dissipated energy seen or measured in practical tests? 

A: The absorbed energy during the real test was calculated according to equation 11. Bearing in to account the correctness of the numerical model calibration, the absorbed energy and dissipated energy were calculated using FEM method. 

  1. Often in the whole paper the term “auxetic core” is mentioned (even in the title), but there is not even one remark on the importance of this auxetic behaviour of the core, on possible influence of the core on the results etc. How much would the results differ is a “standard” cellular formed core would had been used? Either you give a clear explanation on the influence of the “auxetic core” and its special behaviour (if at all) compared to standard cellular cores, or at least the title of the paper must be changed. 

A: The description of the main properties of auxetic materials (structures) has been added in the introduction. However, this study did not emphasize the special influence of the auxetic core on the test results. The type of core is a constant factor in the research. In line with the aim of the study, only the influence of facings type on mechanical properties of beams was investigated. Therefore, at work, little attention is paid to the auxetic core. But, it was necessary to be emphasized, however, that the core is a cellular and auxetic structure, it means with a negative Poisson's ratio. Thus, the beam used is different from traditional sandwich beams with a honeycomb core. Auxetic core cells exhibit a different behavior compared to traditional hexagonal cells. As described in the works [12-14,16], the auxetics increase their width during tension and decrease the width under compression. From this reason, close to the bottom facings, the core was tensiled and minimally expanded. But close to the top facings, the core was compressed and slightly decreased in width. These phenomena do not exist in conventional honeycomb panels. Therefore, auxetic cores enable the formation of honeycomb panels with synclastic surfaces.

Appropriate descriptions have been attached to the text of the work. 

  1. See also above: a possible influence of the PVAc bond line has not been mentioned or considered in the whole paper. Please add relevant comments. The reviewer is sure that a thermoplastic bond line can have influence on the results. 

A: The phenomenon of thermoplasticity mentioned by the Reviewer #2 does not occur in the laboratory conditions in which the test was carried out (constant air humidity 60%, constant air temperature 21°C and constant load speed 10 mm/min). The phenomenon of the thermoplastic behavior of the glue line would be noticeable at temperatures above 90°C or at variable, higher (over 100 mm / min) load speed. For these reasons, thermoplasticity was not included in the calculations.

The appropriate text has been added to the description of the numerical model. 

  1. How often did the failure occur due to damage of the face material, and how often there was damage of the core by shear stresses? Also for the face material: were there only tensile fractures, or also damage by shear stresses? 

A: The crack analysis was supplemented with a description of the problem raised by Reviewer #2.

The significant effect of type of facings material on mechanical properties of multilayered beams could be noted for the crack occurrence for the facings and core structure of tested beams (Fig. 11). For the plywood PL(L), the main crack occurs in the facing material. In this case, the highest local value of the reduced Mises stresses is equal to 32 MPa (Fig.11a). Visible cracks in the cell walls of the core are caused by small stresses not exceeding 3 MPa. For the PL (R) beams, local damage stresses are equal to 26.2 MPa (Fig. 11b). Cracks in facings made of HDF are starting to occur at stresses of approximately 9.2 MPa, while in the case of cardboard facings at stresses of approximately 8.3 MPa (Fig. 11c,d). It should be noted that the damage appeared mainly in the bottom facings. When the above-mentioned local reduced stresses were exceeded, the facings cracked. As a result of these cracks and the widening of the gaps, also there were cracks in the cell walls of the core. This was due to the dominant presence of normal stresses and a small share of tangential stresses. Therefore, the impact of shear stresses on core damage was small. As long as there was no crack in the facings, no cracks in the core were observed.

Jerzy Smardzewski

Reviewer 3 Report

This manuscript investigates the bending behavior of beams of auxetic core and facings of different wood composites. The studies performed in this manuscript are interesting and of important value to the literature. I recommend accepting the manuscript after minor corrections:

  • The authors are recommended to not include many of the results such as Figure 2 and Table 1 in the materials and methods section. I believe they can move some of these results to the results section.
  • The authors should consider the significant figures and standard deviation of the reported numerical values such as Figure 7, Figure 8, Table 1, and Table 3.
  • The authors are recommended to check the whole manuscript for language mistakes and typos.

Author Response

Responses to the Editor and the Reviewers

Dear Editor and the Reviewers,

First of all, we want to thank for offering the authors the opportunity to respond to the reviewers comments. Below, step by step, we present our explanations and changes introduced to the text. All changes (in manuscript) are marked in red.

Responses to Reviewer #3:

We are grateful to the Reviewer for careful reading the manuscript and helpful remarks. Below we respond to the remarks one by one.

  1. This manuscript investigates the bending behavior of beams of auxetic core and facings of different wood composites. The studies performed in this manuscript are interesting and of important value to the literature. I recommend accepting the manuscript after minor corrections:
  1. The authors are recommended to not include many of the results such as Figure 2 and Table 1 in the materials and methods section. I believe they can move some of these results to the results section.

A: The authors consciously decided to include the results of research on the elastic properties of the materials in the methodological part. These type of results had to been presented before the description of the numerical model of beams and the method of its calibration. We believe that this methodology drew the reader's attention to the correct matching of the results of experimental material tests with the result of numerical calibration. The calibration was necessary to estimate true stress and true strain by equations 1-3. According to the authors, transferring this information to the results section would distort the understanding of the idea of calibration of the beam model.

  1. The authors should consider the significant figures and standard deviation of the reported numerical values such as Figure 7, Figure 8, Table 1, and Table 3.

A: An appropriate correction was made. Only in Figure 8, the results with the accuracy of three significant numbers are left. The use of two significant numbers resulted in the loss of differences between the results.

  1. The authors are recommended to check the whole manuscript for language mistakes and typos.

A: Ones again, the text was carefully checked and corrected.

Jerzy Smardzewski

Round 2

Reviewer 2 Report

Most of the comments of the first round of Review have been considered.

There are few left (see attachment).

There are also some new comments needing slight correction in the text.

Author Response

Title: Bending behavior of lightweight wood based sandwich beams with auxetic cellular core

Krzysztof Peliński, Jerzy Smardzewski *

Responses to the Editor and the Reviewers #2

Dear Editor and the Reviewer,

Please let me express our deep thanks for offering ones again the authors the opportunity to prepare responses to the Reviewer comments. Below, step by step, we present our explanations and changes introduced to the text. All changes (in manuscript) are marked in red. The Reviewer's suggestions significantly improved the quality of the manuscript.

     1. … etc.; especially …, … increase of the …

A: Thank you for the suggestion. The correction has been made to the text.

     2. … WoodEpox is not a biomaterial. Comments like this damage the valuable intention to use real biomaterials (materials based on natural resources). This must be changed.

     3. Superform plywood and HDF are classical wood based composites (better term is "wood based panels"). Please change, because it is not correct as it is written.

 A: Thank you for the suggestion. The correction has been made to the text.

"The cognitive goal of the study was to demonstrate the possibility of using wood based materials for facings and cores multilayered honeycomb panels. Besides, the results of the research were to prove that these materials are excellent for the production of beams susceptible to high deflections".

     4. To which extent are cardboard and WoodEpox recycling materials?

A: Thank you for the suggestion. The correction has been made to the text.

"The choice of cardboard and WoodEpox® entailed an attempt to use materials that could be admixed with newly manufactured wood-plastic composites as part of the recycling process".

     5. … based on epoxy resin and a lignin-cellulose mass as filler, …

A: Thank you, this sentence was added to the text.

     6. Thermoplastic behaviour would be rather seen at low load speed, not at high ones.

A: Yes, this is true if we take into account the loads over a long period. In the case of static loads in a short period, the strains in the glue line are determined by the rate at which loads are applied. The paper describes a case of a low rate of load applied over a short period. Therefore, the authors are sure that the phenomenon of thermoplasticity did not occur.

     7. … 85 instead of 85.33

A: Thank you for the suggestion. The correction has been made to the text.

     8. …, but beams with PL(R) facings exhibited values as low as 51 N, which means a reduction by 77%.

A: Thank you for the suggestion. The correction has been made to the text.

 On behalf of all authors, Jerzy Smardzewski
